# Molecular characterization of haemagglutinin genes of influenza B viruses circulating in Ghana during 2016 and 2017

Alhassan Mohammed Yakubu[1,2]*, Nii Ayite Aryee[1], Evelyn Yayra Bonney[3], Erasmus Nikoi Kotey[3,4,5], Joseph Humphrey Kofi Bonney[3,5], Michael R. Wiley[6], Catherine B. Pratt[6], Grace Korkor Ababio[1], Shieley Nimo-Paintsil[5], Naiki Puplampu[5], Seth Attoh[7], Raymond D. Fatchu[7], Edward Owusu Nyarko[2], Anne Fox[5], Chaselynn M. Watters[5], Terrel Sanders[5], Andrew G. Letizia[8], William Kwabena Ampofo[3]

1 Department of Medical Biochemistry, School of Biomedical and Allied Health Sciences, University of Ghana, Accra, Ghana, 2 Public Health Division, 37 Military Hospital, Accra, Ghana, 3 Virology Department, Noguchi Memorial Institute for Medical Research, University of Ghana, Accra, Ghana, 4 Department of Biochemistry, Cell and Molecular Biology, University of Ghana, Accra, Ghana, 5 U.S. Naval Medical Research Unit–No. 3 (NAMRU-3) Ghana Detachment, Accra, Ghana, 6 College of Public Health, University of Nebraska Medical Center, Omaha, Nebraska, United States of America, 7 Pathology Division-Laboratory, 37 Military Hospital, Accra, Ghana, 8 U.S. Naval Medical Research Center (NMRC) Infectious Diseases Directorate, Silver Spring, Maryland, United States of America

* alhassankasul@yahoo.com

**Data Availability Statement:** The manuscript contains the minimal data set by which conclusions were done. Additional data is

## Abstract

Recent reports of haemagglutinin antigen (HA) mismatch between vaccine composition strains and circulating strains, have led to renewed interest in influenza B viruses. Additionally, there are concerns about resistance to neuraminidase inhibitors in new influenza B isolates. To assess the potential impact in Ghana, we characterized the lineages of influenza B viruses that circulated in Ghana between 2016 and 2017 from different regions of the country: Southern, Northern and Central Ghana. Eight representative specimens from the three regions that were positive for influenza B virus by real-time RT-PCR were sequenced and compared to reference genomes from each lineage. A total of eleven amino acids substitutions were detected in the B/Victoria lineage and six in the B/Yamagata lineage. The strains of influenza B viruses were closely related to influenza B/Brisbane/60/2008 and influenza B/Phuket/3073/2013 for the Victoria and Yamagata lineages, respectively. Three main amino acid substitutions (P31S, I117V and R151K) were found in B/Victoria lineages circulating between 2016 and 2017, while one strain of B/Victoria possessed a unique glycosylation site at amino acid position 51 in the HA2 subunit. Two main substitutions (L172Q and M251V) were detected in the HA gene of the B/Yamagata lineage. The U.S. CDC recently reported a deletion sub-group in influenza B virus, but this was not identified among the Ghanaian specimens. Close monitoring of the patterns of influenza B evolution is necessary for the efficient selection of representative viruses for the design and formulation of effective influenza vaccines.

accessible in GenBank under accession numbers MH748708 – MH748715.

**Funding:** The following authors; AGL, CMW and TS were recipients of a grant by the Armed Forces Health Surveillance Division (AFHSD), Global Emerging Infections (GEIS) Branch; PROMIS number P0142_19_N3_08.02. Their URL is: https://health.mil/Military-Health-Topics/Combat-Support/Armed-Forces-Health-Surveillance-Division The funders had no role in the study design, data collection and analysis, decision to publish, or preparation of the manuscript.

**Competing interests:** The authors have declared that no competing interests exist.

## Introduction

Influenza virus causes an annual estimated global infection rate of 5–10% among adults and 20–30% among children along with a persistent concern for epidemics [1]. Presently, seasonal trivalent influenza vaccines are designed to protect against three influenza virus strains; influenza A H1N1pdm09, A H3N2, and typically a circulating B lineage virus (either Yamagata or Victoria) with both B lineages present in quadrivalent vaccines.

Influenza B virus is responsible for about 25% of laboratory documented influenza infections worldwide [2]. This virus has the ability to cause fulminant diseases, such as Reye's syndrome, and can result in severe illness, especially among children. Between 2010 and 2011, influenza B caused about 38% of pediatric influenza deaths in the U.S. Continuous surveillance for influenza viruses to constantly characterize the circulating virus strains is necessary for public health planning and preparedness [3].

Despite the fact that there are two antigenically distinct lineages of influenza B viruses, termed Yamagata and Victoria, co-circulating globally since 1985 [4], only one lineage has been selected for inclusion in current trivalent influenza vaccines. Trivalent vaccines provide limited immunity against strains of the other lineage [5]. In 5 out of 10 influenza seasons (2001 to 2011), the predominant circulating influenza B lineage was different from that chosen for the vaccine [6]. As a result, influenza vaccination campaigns have had limited effectiveness against influenza B during seasons in which a significant proportion of the disease was caused by a strain of influenza B other than the vaccine strain [6].

According to a 2017 World Health Organization (WHO) update, 39 influenza B Victoria (B/Vic) deletion variant viruses have been detected in the United States [7]. These deletion subgroups are likely to impose a future epidemic threat due to the virus ability to escape human immune defense [7].

In sub-Saharan Africa, influenza viruses are responsible for 10% of outpatient acute respiratory cases and 7% of child hospitalizations with acute respiratory infection (ARI) [8]. In Ghana, it is estimated that influenza viruses, in general, were responsible for 9% of medically attended severe acute respiratory illness (SARI) and 18% of influenza-like illness (ILI) among children [9]. Morbidity associated with influenza causes direct economic burdens arising from health-care costs, lost days of work or education, and general social disruption across all age groups [10].

Influenza virus haemagglutinin (HA) and neuraminidase (NA) proteins are the two main glycoproteins on the surface of the virus particle that are involved in the interaction between the host cells and the virus [11]. Influenza B viruses have only one HA and one NA subtype [11]. Reports have recently indicated vaccine mismatch with respect to the influenza B viruses in circulation in parts of United States and China. These variants, currently known as the "deletion sub-group", possess an amino acid deletion at position 162, 163 and/or 164 [7].

Glycosylation of the HA and NA proteins have been shown to impact the virus structure, stability, function, and most importantly virus escape from the host defensive mechanisms [12]. In Ghana, little data other than molecular characterizations of representative strains by the WHO Collaborating Center (CC) on influenza, U.K., is available on influenza B virus infection [13]. Thus, the surveillance activity discussed in this report focused on influenza B virus strains that circulated in Ghana between 2016 and 2017, informing vaccine development and policy makers by updating molecular epidemiological data on influenza B.

In addition to the global health effects of influenza on international public health system in Ghana, influenza also remains high priority respiratory pathogen of U.S. military relevance; respiratory surveillance activities, such as reported in this report, are essential to better understand the transmission dynamics, epidemiologic trends, and emergence of new strains or

circulating variations to guide military force health protection (FHP) measures. The molecular epidemiologic data collected from this activity, in turn, can further inform preventive health measures and be utilized to develop countermeasures to prevent outbreaks and illnesses that could threaten the operational mission readiness of both the U.S. military and the Ghanaian Armed Forces personnel.

## Materials and methods

### Research design

This retrospective molecular epidemiologic analysis utilized archived influenza B positive clinical samples collected during 2016 and 2017 as part of routine public health surveillance activities of the National Influenza Centre (NIC), which is housed in the Virology Department of the Noguchi Memorial Institute for Medical Research, University of Ghana, Accra.

Eight participants (7 males and 1 female) were studied. The reported clinical presentation included fever (88%), cough (75%), sore throat (25%) and difficulty breathing (25%).

Four oropharyngeal/nasopharyngeal swabs from humans were selected for each year (2016 and 2017). These had been previously determined to be Influenza B virus positive by QIAGEN One Step real-time RT-PCR kit (QIAGEN, USA) using the standard U.S. CDC protocol [14]. These samples were selected as representatives from the 3 different zones of the country, Southern, Northern and Central Ghana. Respiratory specimen positive for Influenza B by real-time RT-PCR during subtyping and with a CT <29 were selected for sequencing and further analysis.

### Sequencing of influenza B HA gene

Viral RNA was extracted using the QIAamp viral RNA kit (Qiagen, Hilden, Germany) according to the manufacturer's instructions. RNA extracts were amplified by RT-PCR targeting the influenza B HA genes [14]. Agarose gel electrophoresis was used to confirm the amplicons, which were purified and sequenced on a 3130xl Genetic analyzer (ABI 3130xl, Applied Biosystems, USA).

The HA genes from the selected samples were amplified using the QIAGEN OneStep RT-PCR kit (QIAGEN, USA). The whole HA gene (≈1800bp) was amplified using eight pairs of primers on an ESCO Thermal Cycler (ESCO Microplate Ltd, Singapore). The amplicon sequencing strategy and primer sequences spanning the whole HA gene are displayed in Fig 1 and S1 Table. The reagent formula and protocol for amplification conditions were conducted per the CDC protocol adopted by the NIC and the product was purified [14]. Agarose gel electrophoresis was used to confirm amplicon size. A representative gel image is shown in S1 Fig. Cycle sequencing was performed using the BigDye® Terminator v3.1 Cycle Sequencing kit (Applied Biosystems, USA) per manufacturer's instructions.

### Phylogenetic analysis

The sequences of the eight fragments of the HA gene of the influenza B Victoria and B Yamagata lineages were viewed in Chromas software, version 2.6.2 to identify background noise of the bases called by the genetic analyser. Corresponding sequences to either B Victoria or B Yamagata lineages were edited on the BioEdit platform, version 7.2.5. The fragments of the Influenza B/Victoria and B/Yamagata lineage viruses were assembled to the reference sequences of B/Brisbane/60/2008 and B/Phuket/3073/2013, respectively, using the Geneious software version 10.2.3 [15]. Assembled sequences were aligned with at least one sequence from each continent available on the National Center for Biotechnology Information (NCBI)

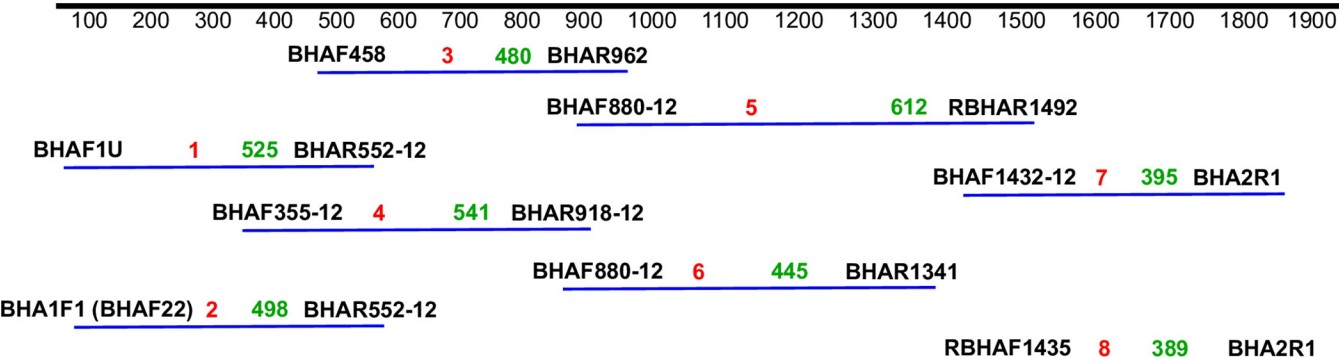

**Fig 1. A schematic of primer locations on the influenza B HA gene.** A schematic representation of the influenza B HA gene showing overlapping fragments and their expected sizes in base pairs. All sixteen primers and their corresponding eight fragments are shown. Red and green colors indicate fragment numbers and expected sizes, respectively.

or the Global Initiative on Sharing All Influenza Data (GISAID) databases. A multiple sequence alignment was carried out using ClustalW in BioEdit with a boostrap replicates of 1000 in line with Edgar [16] (S2 and S3 Figs). Phylogenetic analyses were done using Molecular Evolutionary Genetic Analysis version seven (MEGA7) with the Neighbour-joining model [17]. The phylogenetic tree was constructed and Bootstrap values over 80% have been indicated [18]. The analysis of antigenic drift was performed for both lineages using the WHO vaccine candidate virus sequences for 2018–2019 Influenza season (B/Brisbane/60/2008, the Victoria lineage and B/Phuket/3073/2013, the Yamagata lineage). Sequences generated from this retrospective analysis are accessible in GenBank under accession numbers MH748708 – MH748715 (S2 Table).

## Ethical consideration

This epidemiologic activity was approved by College of Health Sciences Ethical and Protocol Review Committee (CHS-Et/M.3-P 2.7/2017-2018), Noguchi Memorial Institute for Medical Research Institutional Review Board (NMIMR-IRB CPN 062/17-18), and the Naval Medical Research Center Institutional Review Board and Office of Research Administration (NAM-RU3-PJT-21-01) as public health surveillance. A waiver was also sort from Ghana Health Service and approval was given by a letter with reference number GHS/PDH dated Wednesday, December, 18 2019 which clearly stated that this work was purely a public health epidemiologic surveillance and does not require human subject approval from an IRB.

## Results

### Nucleotide changes distinguishing influenza B Victoria from B Yamagata lineages

Targeted sequencing of the full HA genes confirmed the lineage assignment determined by RT-PCR. Four of the samples were from the B Victoria lineage, and four from the B Yamagata lineage. Sequencing further determined that all four of the B Yamagata samples were from Clade 3 of the B Yamagata lineage (Fig 2).

### Analysis of antigenic drift on influenza B Victoria HA gene

After comparing the recently circulating Ghanaian strains from this report with that of the 2018–2019 WHO recommended quadrivalent vaccine candidate virus sequences (B/Brisbane/

```
                          500         510         520         530         540
                    ....|   ....|....|   ....|....|   ....|....|   ....|....|   ....|
B/Brisbane/60/2008       GGATT   TTTCGCAACA   ATGGCTTGGG   CCGTCCCAAA   AAACGACAAA   AACAA
B/Ghana/FS/1688/2016     GGATT   CTTCGCAACA   ATGGCTTGGG   CCGTCCCAAA   AAACGACAAA   AACAA
B/Ghana/FS/1980/2016     GGATT   CTTCGCAACA   ATGGCTTGGG   CCGTCCCAAA   AAACGACAAA   AACAA   B/Vic
B/Ghana/ARI/0005/2017    GGATT   CTTCGCAACA   ATGGCTTGGG   CCGTCCCAAA   AAACGACAAA   AACAA
B/Ghana/ARI/0090/2017    GGATT   CTTCGCAACA   ATGGCTTGGG   CCGTCCCAAA   AAACGACAAA   AACAA

B/Massachusetts/02/2012  GGATT   TTTTGCAACA   ATGGCTTGGG   CTGTCCCAAA   GGACAACAA-   --CAA   B/Yam Clade 2
B/Wisconsin/01/2010      GGATT   TTTCGCAACA   ATGGCTTGGG   CTGTCCCAAA   GGACAACTA-   --CAA
B/Ghana/FS/0730/2016     GGATT   TTTYGCAACA   ATGGCTTGGG   CTGTTCCAAA   GGACAACTA-   --CAA   B/Yam Clade 3
B/Ghana/FS/1912/2016     GGATT   TTTCGCAACA   ATGGCTTGGG   CTGTTCCAAA   GGACAACTA-   --CAA
B/Ghana/FS/0747/2017     GGATT   TTTCGCAACA   ATGGCTTGGG   CTGTTCCAAA   GGACAACTA-   --CAA
B/Ghana/FS/0009/2017     GGATT   TTTYGCAACA   ATGGCTTGGG   CTGTTCCAAA   GGACAACTA-   --CAA

                          550         560         570         580         590
                    ....|   ....|....|   ....|....|   ....|....|   ....|....|   ....|
B/Brisbane/60/2008       AACAG   CAACAAATCC   ATTAACAATA   GAAGTACCAT   ACATTTGTAC   AGAAG
B/Ghana/FS/1688/2016     AACAG   CAACAAATCC   ATTAACAATA   GAAGTACCAT   ACATTTGCAC   AGAAG
B/Ghana/FS/1980/2016     AACAG   CAACAAATCC   ATTAACAATA   GAAGTACCAT   ACATTTGCAC   AGAAG
B/Ghana/ARI/0005/2017    AACAG   CAACAAATCC   ATTAACAATA   GAAGTACCAT   ACATTTGCAC   AGAAG
B/Ghana/ARI/0090/2017    AACAG   CAACAAATCC   ATTAACAATA   GAAGTACCAT   ACATTTGCAC   AGAAG

B/Massachusetts/02/2012  AAATG   CAACGAACCC   ATTAACAGTA   GAAGTACCAT   ACATTTGTGC   AGAAG
B/Wisconsin/01/2010      AAATG   CAACGAACCC   ACTAACAGTA   GAAGTACCAT   ACATTTGTAC   AGAAG
B/Ghana/FS/0730/2016     AAATG   CAACGAACCC   ACAAACAGTG   GAAGTACCAT   ACATTTGTAC   AGAAG
B/Ghana/FS/1912/2016     AAATG   CAACAAACCC   ACAAACAGTG   GAAGTACCAT   ACATTTGTAC   AGAAG
B/Ghana/FS/0747/2017     AAATG   CAACGAACCC   ACAAACAGTG   GAAGTACCAT   ACATTTGTAC   AGAAG
B/Ghana/FS/0009/2017     AAATG   CAACGAACCC   ACAAACAGTG   GAAGTACCAT   ACATTTGTAC   AGAAG
```

**Fig 2. Lineage specific markers of influenza B HA gene.** Multiple sequence alignment was carried out using ClustalW in BioEdit with a boostrap replicates of 1000 in line with Edgar [16]. Influenza B/Brisbane/60/2008 was used as the reference sequence for B/Victoria lineages while B/Wisconsin/1/2010, Clade 3 and B/Massachusetts/2/2012, Clade 2 were used as the reference sequence for B Yamagata lineages. Influenza B virus lineage-specific markers (nts 522, 540–542, 548, 549, 555, 558 and 568) are shown in yellow, whereas the clade specific markers (nts 538, 562 and 589) have been highlighted as green.

60/08), a total of eleven amino acid substitutions were detected in the HA gene (Fig 3). Out of these, five amino acid substitutions were detected in the HA1, with six in the HA2 (S3 Table). All four Ghanaian specimens had P31S and I117V mutations in HA1 and R151K mutation in HA2. Amino acid substitutions varied per sample from three to six. The two Ghanaian reference sequences (B/Ghana/DILI-16-1091/2016 and B/Ghana/FS-16-1620/2016) added had four amino acid substitutions at the same position as compared to the sequences analyzed in this report. Three samples also had N53T, E82D, I92K, E97K and S103A amino acid substitutions in the HA2. None of the sequences from our analysis were found to be among the deletion sub-group (amino acid deletions at position 162, 163 and/or 164). Details of the mutations detected in the individual specimens are shown in (S3 Table).

## Analysis of antigenic drift on influenza B Yamagata HA gene

The influenza B Yamagata lineage HA gene from our data analysis had fewer amino acid substitutions compared to Victoria lineage (Fig 4). When the sequences from this report were compared with that of the 2018–2019 WHO quadrivalent vaccine candidate, the Yamagata strain sequence (B/Phuket/3073/2013), our influenza B/Ghana/FS/1912/2016 and influenza B/Ghana/FS/0747/2017 samples each had four amino acid substitutions while the other sequences had three amino acid substitutions. All the B Yamagata lineage sequences identified, including the two reference Ghanaian sequences, had L172Q and M251V amino acid

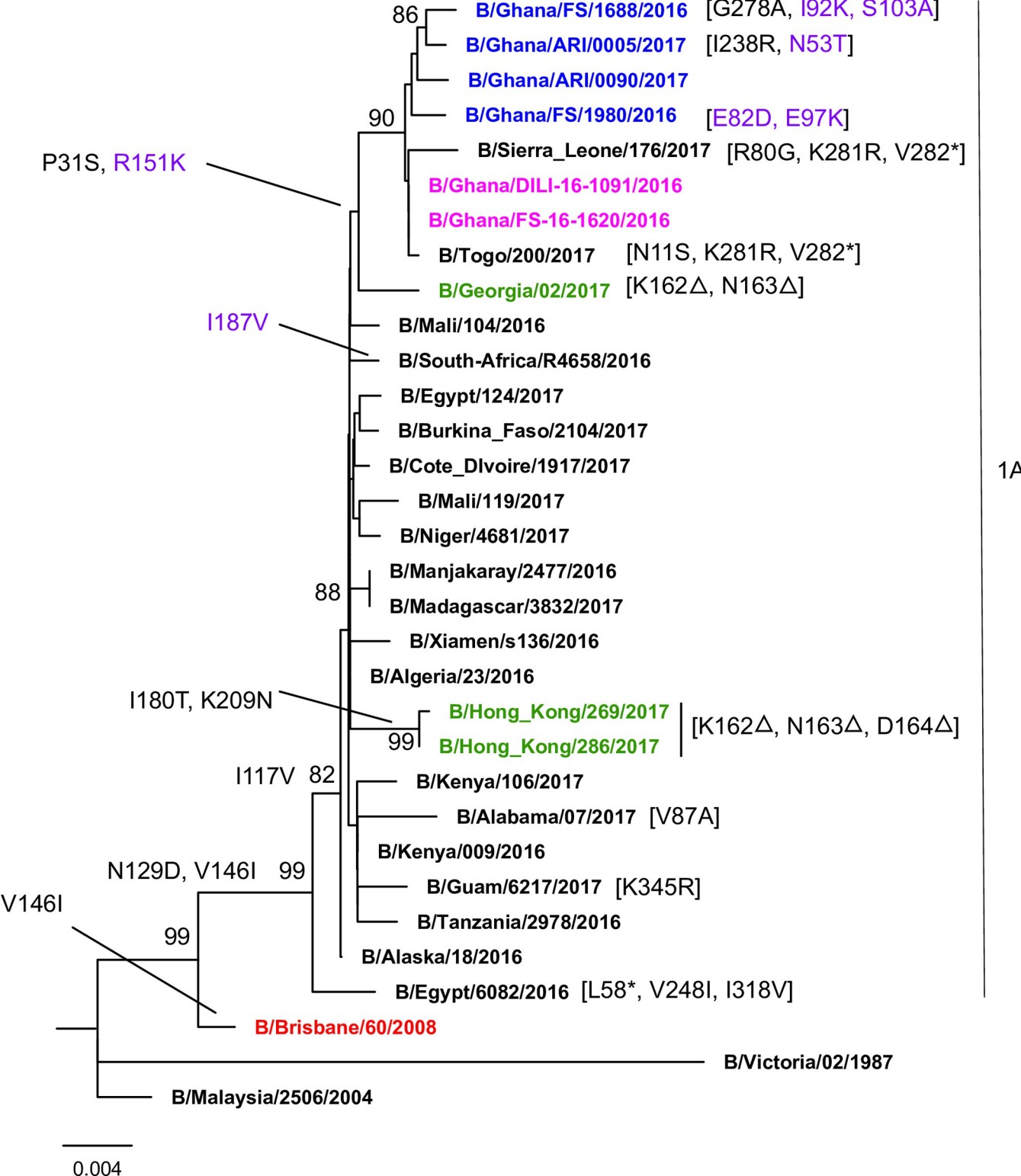

**Fig 3. Phylogenetic analysis of influenza B Victoria lineage using HA genes.** Bootstrap values over 80% are indicated on the tree. Red represents the WHO vaccine candidate virus genome, pink represents reference Ghanaian specimens sequenced at the Francis Crick Institute, blue represents the sequences obtained from our retrospective analysis, green represents the deletion sub-group, Amino acid changes in black represent those within HA1, with violet representing changes in the HA2.

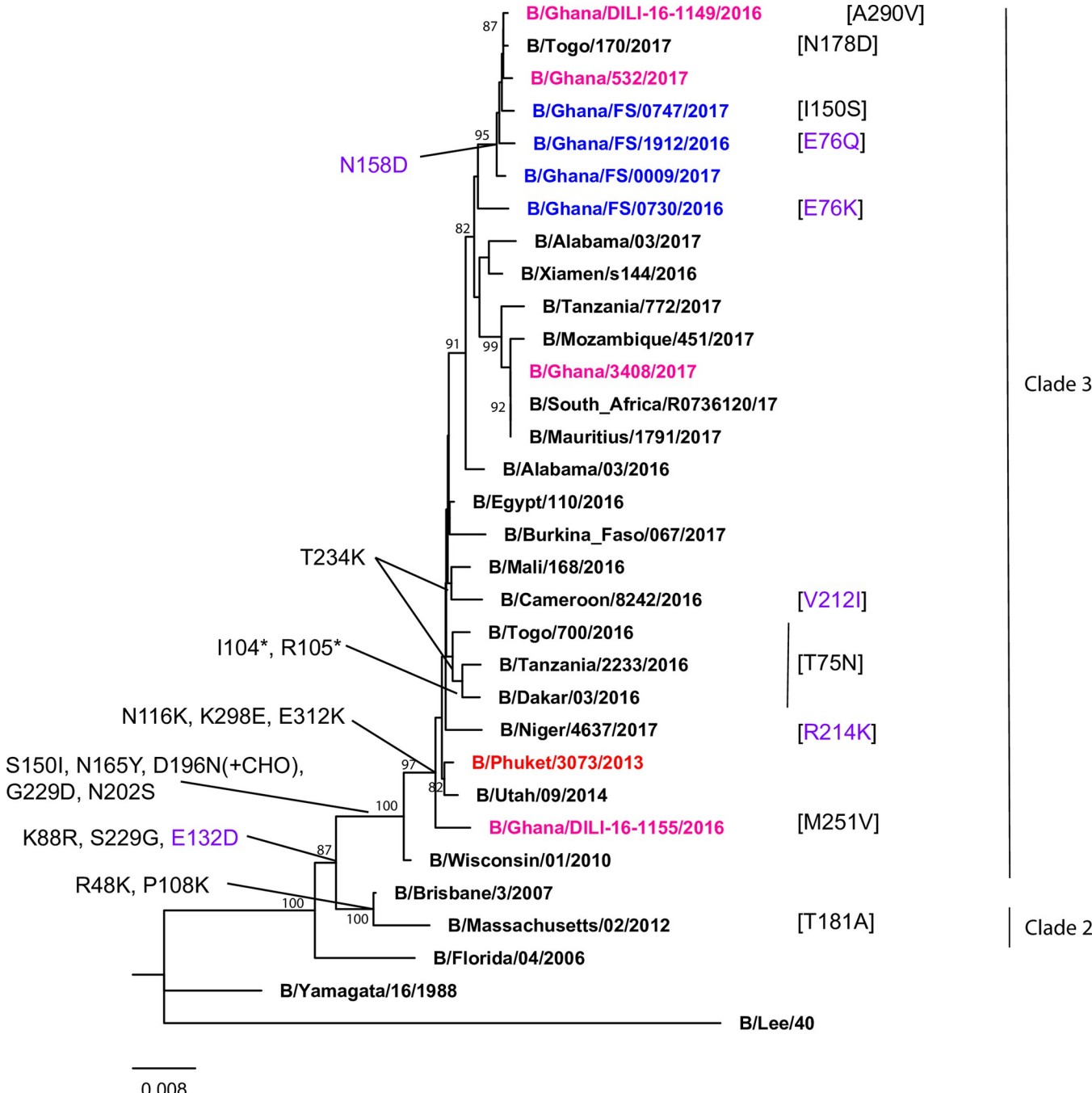

**Fig 4. Phylogenetic analysis of influenza B Yamagata lineage using the HA genes.** Bootstrap values over 80% are indicated on the tree. Red represents the WHO vaccine candidate virus genome, pink represents reference Ghanaian specimens sequenced at the Francis Crick Institute, blue represents the sequences we identified, green represents the deletion sub-group, Amino acid changes in black represent those within HA1, with violet representing changes in the HA2.

substitutions within the HA1 gene. Only one specimen, Influenza B/Ghana/FS/0747/2017 had an additional I150S amino acid substitution within the HA1 gene. All of the sequences identified, and Ghanaian reference sequences (B/Ghana/DILI-16-11149-2016 and B/Ghana/532/2017) showed diversity at position I/T198T of the HA1 gene compared to the vaccine strain. A majority (83%) of the sequences in this report, including the two Ghanaian reference

sequences, had N158D amino acid substitution within the HA2 gene. In addition, two different amino acid substitutions were detected at the same position on the HA2 gene. Influenza B/Ghana/FS/0730/2016 had an E72K amino acid substitution while B/Ghana/FS/1912/2016 had an E72Q amino acid substitution. There was no detection of amino acid deletions at positions 162, 163 and 164 among the sequences from our analysis. The details of mutations in our samples in relation to the reference sequence B/Phuket/3073/2013 are shown in S4 Table.

## Inferring glycosylation

The analysis of glycosylation of the influenza B Victoria (S5 Table) revealed that one of the specimens (B/Ghana/ARI/0005/2017) gained a glycosylation site at position 51 of the HA2 gene relative to the vaccine sequence (B/Brisbane/60/2008). All the sequences analyzed, including the reference strains, demonstrated gain of glycosylation site at position 196 of the HA1 gene of influenza B Yamagata lineage, but absent in the vaccine candidate virus sequence due to amino acid degeneracy (S6 Table) [19]. Detailed presentations of amino acid alignment showing the potential N-glycosylation sites identified for both B Victoria and B Yamagata lineages are shown in S2 and S3 Figs.

## Discussion

Phylogenetic analyses of both influenza B Victoria and Yamagata lineages (Figs 3 and 4, respectively) show that the influenza B Victoria sequences from our findings cluster together and within the same clade as the vaccine virus sequence (B/Brisbane/60/2008). Three of the B/Yamagata sequences identified clustered together with the reference Ghanaian sequences and the fourth sequence clustered separately. These clustering patterns are similar to previous reports of Ghanaian sequences between 2016 and 2017 flu season [20].

Molecular characterization of influenza B is vital in the assessment of similarities and differences between vaccine candidate viruses and circulating viruses. This study described the influenza B strains that circulated in Ghana between 2016 and 2017 by sequencing the HA genes. Eight specimens of B Victoria and B Yamagata lineages of influenza B from multiple regions in Ghana were studied. Characterization helps inform the World Health Organization in formulating an effective vaccine for the subsequent influenza seasons. However, due to mutations within the influenza viral genome, achieving this goal in challenging and this study helps describe those differences.

The nucleotide changes that distinguish influenza B Victoria from that of influenza B Yamagata lineages from this study support those reported by Arvia et al. [21], except nucleotide positions 540–542 and 538, which are different. Antigenic variation in influenza B virus is mostly caused by amino acid substitutions at four major antigenic epitopes (120-loop, 150-loop, 160-loop and 190-helix), as identified in previous studies [22]. In this study, six amino acid substitutions were observed in the HA1 region, relative to the Influenza B Victoria lineage sequence B/Brisbane/60/2008 (the 2016–2017 trivalent and the 2017–2018 quadrivalent influenza vaccine component) [23]. These amino acid substitutions are P31S, I117V, N129D, G278A, K281R, and I283R. However, only two of these substitutions (I117V and N129D) were located near the antigenic sites, specifically around the conserved 120-loop of the HA1 region. Substitution of N with D at residue 129 have been reported to affect vaccine correspondence to a mutant virus [24]. The amino acid substitutions P31S, I117Vand N129D that we identified are similar to the Ghanaian reference sequences included in these analyses. All the sequences from this study, including the reference sequences from Ghana, had the P31S and I117V substitutions. Of note, each sequence from this study had a unique amino acid substitution. For example, influenza B/Ghana/FS/1688/2016 had a G278A amino acid

substitution relative to the vaccine component, while Influenza B/Ghana/ARI/0005/2017 had an I283R amino acid substitution.

In the HA2 region, the amino acid substitution R151K (R498K of HA) was similar to a mutation observed in the reference Ghanaian sequences. In all, five amino acid substitutions were observed in all four sequences. Four other amino acid substitutions were identified to be unique to only three sequences from this study: Influenza B/Ghana/ARI/0005/2017 had a N53T in the HA2 (N400T), influenza B/Ghana/FS/1980/2016 was detected to have two amino acid substitutions at residues E82D and E97R in the HA2 (E429D and E444R) and influenza B/Ghana/ FS/1688/2016 to have S103A amino acid substitution in the HA2 (S450A). These substitutions may have an effect on the fusion ability of influenza B HA2 domain due to the change from a non-charged polar amino acid (serine) to a non-charged non-polar amino acid (alanine). This, unless otherwise confirmed to have no effect, may be a potential cause of vaccine mismatch, and hence requires further phenotypic assays for complete characterization.

For the influenza B Yamagata sequences, all the four viruses sequenced, in addition to the Ghanaian reference viruses sequenced by the WHO CC fell into a group defined by the HA substitutions L172Q and M251V in HA1 and N158D in HA2 (N504D) from B/Phuket/3073/ 2013 in clade 3 (the B/Wisconsin/1/201 0-B/Phuket/3073/2013 clade) except influenza B/ Ghana/FS/0730/2016, which did not show any amino acid change at N158 in HA2 (N504). Apart from the above, there are parallel amino acid substitutions observed in three of the sequences from this study that were not found in the Ghanaian reference sequences. These amino acid substitutions were found in influenza B/Ghana/FS/0747/2017 at I150S of the HA1 domain while the other two are found in influenza B/Ghana/FS/0730/2016 and influenza B/ Ghana/FS/1912/2016 at residues E76K and E76Q in the HA2 (E422K and E422Q), respectively. Amino acid degeneracy was also observed at I/T198T relative to the WHO vaccine candidate (influenza B/Phuket/3073/2013).

In this study, the glycosylation pattern revealed by influenza B Victoria lineages from the Ghanaian origin were similar to influenza B/Brisbane/60/2008 except for influenza B/Ghana/ ARI/0005/2017, which showed a gain in glycosylation site at position 51 of the HA2 (398) domain [25]. This gain of a glycosylation site is not likely to unmask new antigenic sites because this site is not reported to be an epitope on the HA protein. However, previous evidence for the importance of carbohydrates in modulating antigenicity has been presented by selection of a mutant HA (D197N), which resulted in a new glycosylation site that prevented antibody binding and viral neutralization [26].

The glycosylation pattern of influenza B Yamagata lineages from this study was the same in the reference Ghanaian sequences, except for a gain of glycosylation at position 196 as compared with the vaccine virus sequences (influenza B/Phuket/3073/2013). This gain of a glycosylation site of the Yamagata lineage is likely to unmask a new antigenic site because this site is located within the epitope 190—helix of the HA1 gene. A single mutation of A196T can lead to the creation of a new potential glycosylation site at HA1 (194–196) domain rendering the virus epidemic [24]. None of the deletion variants reported by the WHO CC in 2017 were identified among the Ghanaian sequences from this study [7].

## Conclusions

Due to the nature of respiratory pathogens like influenza being easily transmissible and the detrimental impact to overall productivity and mission readiness, the U.S military prioritizes ongoing respiratory pathogen surveillance and genomic sequencing efforts throughout the U. S. Africa Command area of responsibility. B Yamagata and B Victoria sequences analyzed from Ghana demonstrated only minor drifts, but were genetically like the vaccine reference

strains, B/Phuket/3073/2013 and B/ Brisbane/60/2008, respectively. Furthermore, the recent deletion sub-group in Influenza B virus reported by the CDC was not identified among the specimens [7, 27].

Of the four samples that were of the Victoria lineage, 3 notable substitutions (P31S, I117V and R151K) were identified. One of the Influenza B Victoria sequences (B/Ghana/ARI/0005/2017) analyzed had gained a unique glycosylation site at amino acid position 51 in the HA2 subunit, and this was absent from all other sequences reported. The other four samples that belonged to the Yamagata, lineage, had only 2 notable substitutions (L172Q and M251V) were identified.

This retrospective analysis is the first to optimize an existing influenza B sequencing protocol to sequence influenza B viruses in Ghana while providing valuable public health data on influenza strains that are in circulation.

## Supporting information

**S1 Fig. A sample gel image.**
(PDF)

**S2 Fig. HA amino acid alignment for influenza B Victoria-lineage viruses.**
(PDF)

**S3 Fig. HA amino acid alignment for influenza B Yamagata-lineage.**
(PDF)

**S1 Table. Details of primers used for this research.**
(PDF)

**S2 Table. Accession numbers of influenza B sequences in NCBI influenza Virus Resource and GISAID used for the phylogenetic reconstruction of HA genes.**
(PDF)

**S3 Table. Amino acid substitutions in the HA genes of the study sequences, compared to the Influenza B Victoria reference strain B/Brisbane/60/2008.**
(PDF)

**S4 Table. Amino acid substitutions in the HA genes of the study sequences, compared to the influenza B Yamagata reference strain B/Phuket/3073/2013.**
(PDF)

**S5 Table. Potential glycosylation sites of influenza B Victoria HA genes.**
(PDF)

**S6 Table. Potential glycosylation sites of influenza B Yamagata HA genes.**
(PDF)

**S1 Raw images. Uncropped gel images.**
(PDF)

## Acknowledgments

We are grateful to the fellows and colleagues of the National Influenza Centre (NIC), Noguchi Memorial Institute for Medical Research, for granting us space and equipment for the work.

## Author Contributions

**Conceptualization:** Alhassan Mohammed Yakubu, William Kwabena Ampofo.

**Formal analysis:** Alhassan Mohammed Yakubu, Evelyn Yayra Bonney, Erasmus Nikoi Kotey.

**Investigation:** Alhassan Mohammed Yakubu.

**Methodology:** Alhassan Mohammed Yakubu.

**Writing – original draft:** Alhassan Mohammed Yakubu, Erasmus Nikoi Kotey.

**Writing – review & editing:** Nii Ayite Aryee, Evelyn Yayra Bonney, Joseph Humphrey Kofi Bonney, Michael R. Wiley, Catherine B. Pratt, Grace Korkor Ababio, Shieley Nimo-Paintsil, Naiki Puplampu, Seth Attoh, Raymond D. Fatchu, Edward Owusu Nyarko, Anne Fox, Chaselynn M. Watters, Terrel Sanders, Andrew G. Letizia.

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
