## [Decision Letter · Decision Letter 0]

30 Mar 2022

PONE-D-21-23774Molecular characterization of haemagglutinin genes of influenza B viruses circulating in Ghana during 2016 and 2017PLOS ONE

Dear Dr. Yakubu,

Thank you for submitting your manuscript to PLOS ONE. After careful consideration, we feel that it has merit but does not fully meet PLOS ONE’s publication criteria as it currently stands. Therefore, we invite you to submit a revised version of the manuscript that addresses the points raised during the review process.

 In particular, you need to address reviewer 1's comments about the low number of sequences obtained. Furthermore, addressing all of the other major/minor points will improve the readability of your paper.

We look forward to receiving your revised manuscript.

Kind regards,

Brian M. Ward, Ph.D.

Academic Editor

PLOS ONE

A clean copy of the edited manuscript (uploaded as the new *manuscript* file).

“The work was funded by the Armed Forces Health Surveillance Division (AFHSD), Global Emerging Infections Surveillance (GEIS) Branch; PROMIS number P0142_19_N3_08.02.”

Reviewers' comments:

Reviewer's Responses to Questions

**Comments to the Author**

1. Is the manuscript technically sound, and do the data support the conclusions?

Reviewer #1: Partly

Reviewer #2: Yes

2. Has the statistical analysis been performed appropriately and rigorously? 

Reviewer #1: I Don't Know

Reviewer #2: Yes

3. Have the authors made all data underlying the findings in their manuscript fully available?

Reviewer #1: Yes

Reviewer #2: Yes

4. Is the manuscript presented in an intelligible fashion and written in standard English?

Reviewer #1: Yes

Reviewer #2: Yes

5. Review Comments to the Author

Reviewer #1: The authors characterized the lineages of influenza B viruses that circulated in Ghana between 2016 and 2017. Eight sequences of the complete HA genes (4:Yamagata lineage, 4: Victoria lineage) were included for the phylogenetic analysis. The authors reported eleven amino acids substitutions detected in the B/Victoria lineage and six in the B/Yamagata lineage. However, some issues should be revised to strengthen the manuscript.

Major points:

- The main concern of this manuscript is whether these samples represent the characteristic of influenza B viruses circulating in Ghana. The eight sequences seem a relatively low number for the results obtained from the targeted sequencing (Sanger sequencing), not the whole genome sequencing.

- Are these the only eight sequences positive for influenza B RT-PCR between 2016-2017 from the archive of the study sites? The sample numbers sent to the study sites during 2016-2017 and the samples positive for influenza B viruses during 2016-2017 should be included to calculate the rate of detection. There are no precise inclusion criteria for selecting these eight samples as the representative samples. This information should be included in the materials and methods section.

- The order of figure legends is confusing, starting from Fig. 1, Fig. 4, Fig. 2, Fig. 3.

- Materials and Methods: the method/program used for potential glycosylation analysis should be included.

- There is inconsistent information regarding the analysis of antigenic drift. The materials and methods section stated that the analysis was performed for both lineages using the WHO vaccine candidate virus sequences for the 2018-2019 Influenza season. However, the results section mentioned that the sequences were compared with that of the 2016-2017 WHO trivalent and quadrivalent vaccine candidate.

Minor points:

- Materials and Methods: Line 116-117: These had been previously determined to be Influenza B virus positive by real-time RT-PCR using the standard U.S. CDC protocol [15]. -> If the name of the real-time RT-PCR test kit is available, it should be included.

- Results: The title that states “Nucleotide changes distinguishing influenza B Victoria from Yamagata lineages” provided no new information. It should be revised to present other important points of the study.

- Results: Line 174: Whole genome amplicon sequencing of the HA genes… should be revised to Targeted sequencing of the full HA genes...

- Results: S1 Figure should include the size label of the 100 bp marker.

Reviewer #2: The paper is well written and addresses an important issue such as the choice of the best vaccine component for influenza vaccines. In particular, the problem of choosing component B was addressed.

The samples that have been analyzed are few and this could represent a limit. However, the aim of the work was to identify a path to improve the choice of influenza B vaccine components and not the description of the circulation of influenza viruses in a specific area and in a specific period.

Please make the following changes:

1. In the paper “influenza” sometimes it is written with a capital letter, other times with a lowercase. It is necessary to standardize the way of writing.

2. In the paper sometimes it is reported B/Victoria and B/Yamagata, other times B Victoria and B Yamagata (with or without /). Also, in this case it is necessary to standardize the way of writing

3. It is unclear whether the samples are collected from 2016-17 or during 2016 and 2017 influenza seasons (line 110)

6. PLOS authors have the option to publish the peer review history of their article (what does this mean?). If published, this will include your full peer review and any attached files.

Reviewer #1: No

Reviewer #2: No

---

## [Author Response · Author response to Decision Letter 0]

27 May 2022

Academic Editor

1- Inappropriate funding declaration in manuscript

Corrections made:

All funding-related texts have been removed from the manuscript [Lines 382-384]. 

Kindly consider replacing the funding statement “The funders had no role in study design, data collection and analysis, decision to publish, or preparation of the manuscript.” With the revised funding statement below: 

Revised Funding Statement: “The work was funded by the Armed Forces Health Surveillance Division (AFHSD), Global Emerging Infections Surveillance (GEIS) Branch; PROMIS number P0142_19_N3_08.02.

2- Request for original uncropped and unadjusted images underlying all blot or gel results report in the submission’s figures or supporting information files.

Response:

All original uncropped and unadjusted images have been resubmitted.

Reviewer #1

Major points

1- Are these the only eight sequences positive for influenza B RT-PCR between 2016-2017 from the archive of the study sites? The sample numbers sent to the study sites during 2016-2017 and the samples positive for influenza B viruses during 2016-2017 should be included to calculate the rate of detection. There are no precise inclusion criteria for selecting these eight samples as the representative samples. This information should be included in the materials and methods section.

Response:

No, the eight samples assessed were just a few and they were selected in line with the aim of the study“to identify a path to improve the choice of influenza B vaccine components” and not the description of the circulation of influenza viruses in Ghanain the specific period.

These samples were selected as representatives from the 3 different zones of the country, Southern, Northern and Central Ghana. Specimen positive for Influenza B by real-time RT-PCR during subtyping and with a CT <29 were selected for sequencing and further analysis. Mostly considered because of their strong Ct values upon subtyping and have been mentioned in the materials and methods section [Lines 119-122].

2- The order of figure legends is confusing, starting from Fig. 1, Fig. 4, Fig. 2, Fig. 3.

Response:

 The figure legends have been rearranged from Fig. 1, Fig. 2, Fig. 3 and Fig. 4 [Lines 168-179].

3- Materials and Methods: the method/program used for potential glycosylation analysis should be included.

Response:

We only inferred N-glycosylation sites based on the knowledge of the “glycosylationable” amino acid N-X-S/T sequon [Lines 252-260].

4- There is inconsistent information regarding the analysis of antigenic drift. The materials and methods section stated that the analysis was performed for both lineages using the WHO vaccine candidate virus sequences for the 2018-2019 Influenza season. However, the results section mentioned that the sequences were compared with that of the 2016-2017 WHO trivalent and quadrivalent vaccine candidate.

Response:

The 2016-2017 WHO trivalent and quadrivalent vaccine candidates were erroneously written. This has been duly corrected at the results section [Lines 206-207 and 229].

Minor points

1- Materials and Methods: Line 116-117: These had been previously determined to be Influenza B virus positive by real-time RT-PCR using the standard U.S. CDC protocol [15]. -> If the name of the real-time RT-PCR test kit is available, it should be included.

Response:

 ‘QIAGEN One Step RT-PCR kit (QIAGEN, USA)’ is inserted in lines 117 and 118.

2- Results: The title that states “Nucleotide changes distinguishing influenza B Victoria from Yamagata lineages” provided no new information. It should be revised to present other important points of the study.

Response:

The new information with respect to the above title is shown in this statement ‘The nucleotide changes that distinguish influenza B Victoria from that of influenza B Yamagata lineages from this study support those reported by Arvia et al. (2014), except nucleotide positions 540-542 and 538, which are different’ [lines 287 to 289].

3- Results: Line 174: Whole genome amplicon sequencing of the HA genes… should be revised to Targeted sequencing of the full HA genes...

Response:

Whole genome amplicon sequencing of the HA genes… has been replaced with ‘Targeted sequencing of the full HA genes….. as recommended [Line 193].

4- Results: S1 Figure should include the size label of the 100 bp marker.

Response:

The 100 bp marker (Ladder) has been labeled appropriately in S1 Figure.

Reviewer #2

1. In the paper “influenza” sometimes it is written with a capital letter, other times with a lowercase. It is necessary to standardize the way of writing.

Response:

Writing of ‘influenza’ has been duly standardized within the paper except when ‘influenza’ is beginning with a sentence.

2. In the paper sometimes it is reported B/Victoria and B/Yamagata, other times B Victoria and B Yamagata (with or without /). Also, in this case it is necessary to standardize the way of writing

Response:

Writing of ‘B Victoria and B Yamagata’ have been duly standardized within the paper.

3. It is unclear whether the samples are collected from 2016-17 or during 2016 and 2017 influenza seasons (line 110)

Response:

Samples were collected during 2016 and 2017 influenza seasons. Revisions have been made to reflect the change – “This retrospective molecular epidemiologic analysis utilized archived influenza B positive clinical samples collected during 2016 and 2017 as part of routine public health surveillance activities of the National Influenza Centre (NIC), which is housed in the Virology Department of the Noguchi Memorial Institute for Medical Research, University of Ghana, Accra.” [Lines 109-112].

---

## [Decision Letter · Decision Letter 1]

29 Jun 2022

Molecular characterization of haemagglutinin genes of influenza B viruses circulating in Ghana during 2016 and 2017

PONE-D-21-23774R1

Dear Dr. Yakubu,

We’re pleased to inform you that your manuscript has been judged scientifically suitable for publication and will be formally accepted for publication once it meets all outstanding technical requirements.

Kind regards,

Brian M. Ward, Ph.D.

Academic Editor

PLOS ONE

Additional Editor Comments (optional):

Reviewers' comments:

Reviewer's Responses to Questions

**Comments to the Author**

1. If the authors have adequately addressed your comments raised in a previous round of review and you feel that this manuscript is now acceptable for publication, you may indicate that here to bypass the “Comments to the Author” section, enter your conflict of interest statement in the “Confidential to Editor” section, and submit your "Accept" recommendation.

Reviewer #1: (No Response)

Reviewer #2: All comments have been addressed

2. Is the manuscript technically sound, and do the data support the conclusions?

Reviewer #1: Yes

Reviewer #2: Yes

3. Has the statistical analysis been performed appropriately and rigorously? 

Reviewer #1: I Don't Know

Reviewer #2: Yes

4. Have the authors made all data underlying the findings in their manuscript fully available?

Reviewer #1: Yes

Reviewer #2: Yes

5. Is the manuscript presented in an intelligible fashion and written in standard English?

Reviewer #1: Yes

Reviewer #2: Yes

6. Review Comments to the Author

Reviewer #1: On these comments:

1- Are these the only eight sequences positive for influenza B RT-PCR between 2016-

2017 from the archive of the study sites? The sample numbers sent to the study sites

during 2016-2017 and the samples positive for influenza B viruses during 2016-2017

should be included to calculate the rate of detection. There are no precise inclusion

criteria for selecting these eight samples as the representative samples. This

information should be included in the materials and methods section.

Response:

No, the eight samples assessed were just a few and they were selected in line with the

aim of the study“to identify a path to improve the choice of influenza B vaccine

components” and not the description of the circulation of influenza viruses in Ghanain

the specific period.

These samples were selected as representatives from the 3 different zones of the

country, Southern, Northern and Central Ghana. Specimen positive for Influenza B by

real-time RT-PCR during subtyping and with a CT <29 were selected for sequencing

and further analysis. Mostly considered because of their strong Ct values upon

subtyping and have been mentioned in the materials and methods section [Lines 119-

122].

>>Reviewer's comment on the author's response:

Thank you for the explanation. The current title/abstract gives the impression of the aim to characterize the circulating strains during 2016-2017.

To clarify that the eight samples assessed were the representative samples with the aim to identify a path to improve the choice of influenza B vaccine components, the abstract should be revised to state this point.

For example,

" To assess the potential impact in Ghana, we characterized the lineages of influenza B viruses that circulated in Ghana between 2016 and 2017 from different regions of the country: Southern, Northern, and Central Ghana.

Eight representative specimens from the three regions that were positive for influenza B virus by real-time RT-PCR were sequenced and compared to reference genomes from each lineage."

2- The order of figure legends is confusing, starting from Fig. 1, Fig. 4, Fig. 2, Fig. 3.

Response:

The figure legends have been rearranged from Fig. 1, Fig. 2, Fig. 3 and Fig. 4 [Lines

168-179].

>>Reviewer's comment on the author's response:

- The current figures 2 and 3 should still be in the result section but be renumbered to be figures 3 and 4, while the current figure 4 should be renumbered to be figure 2. Please adjust the order of the attached figures accordingly and remove the citing (Fig.2 and 3) from the Method section-Phylogenetic analysis.

Reviewer #2: (No Response)

7. PLOS authors have the option to publish the peer review history of their article (what does this mean?). If published, this will include your full peer review and any attached files.

Reviewer #1: No

Reviewer #2: No

---

## [Editor Report · Acceptance letter]

2 Sep 2022

PONE-D-21-23774R1 

Molecular characterization of haemagglutinin genes of influenza B viruses circulating in Ghana during 2016 and 2017 

Dear Dr. Yakubu:

I'm pleased to inform you that your manuscript has been deemed suitable for publication in PLOS ONE. Congratulations! Your manuscript is now with our production department. 

Kind regards, 

on behalf of

Dr. Brian M. Ward 

Academic Editor

PLOS ONE